# HER2-Low Status Is Not Accurate in Breast Cancer Core Needle Biopsy Samples: An Analysis of 5610 Consecutive Patients

**DOI:** 10.3390/cancers14246200

**Published:** 2022-12-15

**Authors:** Yujie Lu, Siji Zhu, Yiwei Tong, Xiaochun Fei, Wu Jiang, Kunwei Shen, Xiaosong Chen

**Affiliations:** 1Department of General Surgery, Comprehensive Breast Health Center, Ruijin Hospital, Shanghai Jiao Tong University School of Medicine, Shanghai 200025, China; 2Comprehensive Breast Health Center, Department of Pathology, Ruijin Hospital, Shanghai Jiao Tong University School of Medicine, Shanghai 200025, China; 3Department of Thyroid and Breast Surgery, Yancheng Chinese Medicine Hospital, Yancheng 224001, China

**Keywords:** breast cancer, HER2-Low, concordance, core needle biopsy, surgical excision samples

## Abstract

**Simple Summary:**

Novel anti-HER2 antibody–drug conjugates showed convincing efficacy in HER2-Low breast cancer patients. We aimed to investigate the accuracy of core needle biopsy (CNB) in diagnosing HER2-Low status. We found a low concordance rate of HER2-Low status between CNB and surgical excision specimen (SES) samples in early-stage HER2-Negative patients. In tumors identified as HER2-0 by CNB, 50.3% expressed HER2 at any level at SES samples. Our research confirmed the necessity of retesting HER2-Low status in SES samples to guide precise anti-HER2 ADCs therapy.

**Abstract:**

Background: HER2-Low status is found in approximately half of breast cancer patients and shows potential benefits from novel antibody–drug conjugates (ADCs). Data on the accuracy of HER2-Low status between core needle biopsy (CNB) and surgical excision specimen (SES) samples are lacking. We aimed to investigate the accuracy of HER2-Low status diagnosis between CNB and SES samples. Methods: Consecutive early-stage breast cancer patients who underwent surgery from January 2009 to March 2022 with paired CNB and SES samples were retrospectively reviewed. HER2-Low was defined as IHC 1+ or IHC2+ and FISH-negative. Concordance rates were analyzed by the Kappa test. Further clinicopathological characteristics were compared among different HER2 status and their changes. Results: A total of 5610 patients were included, of whom 3209 (57.2%) and 3320 (59.2%) had HER2-Low status in CNB and SES samples, respectively. The concordance rate of HER2 status in the whole population was 82.37% (Kappa = 0.684, *p* < 0.001), and was 76.87% in the HER2-Negative patients (Kappa = 0.372, *p* < 0.001). Among 1066 HER2-0 cases by CNB, 530 patients were classified as HER2-Low tumors. On the contrary, in 3209 patients with HER2-Low tumor by CNB, 387 were scored as HER2-0 on the SES samples. ER-negative or Ki67 high expression tumor by CNB had a high concordance rate of HER2-Low status. Conclusions: A relatively low concordance rate was found when evaluating HER2-Low status between CNB and SES samples in HER2-Negative breast cancer patients, indicating the necessity of retesting HER2 low status at surgery, which may guide further therapy in the era of anti-HER2 ADCs.

## 1. Introduction

Breast cancer is the most commonly diagnosed malignancy in women worldwide [1], with a high heterogeneity in its biology, clinical features and treatment sensitivity. As an exemplar of precision medicine, human epidermal growth factor receptor-2 (HER2)-targeted therapies have greatly improved the prognosis of HER2-Positive breast cancer patients; however, they have failed to do so in HER2-Negative patients according to several pivotal trials [2]. Recently, the increasing efficacy of novel antibody–drug conjugates (ADCs) in patients with low HER2 protein expression (defined as immunohistochemistry (IHC) 1+ or 2+ without ERBB2 gene amplification by fluorescence in situ hybridization (FISH)) has had a considerable influence in the field of targeted therapy [3,4,5]. These patients are classified as having HER2-Low breast cancer and have become the focus of new clinical and translational research. HER2-Low breast cancer accounts for 40–50% of all breast cancer cases [2] and is considered a distinct biological and clinical subtype of breast cancer [6]. Two phase 3 clinical trials are currently evaluating the efficacy of trastuzumab deruxtecan (T-Dxd, DS-8201) in this dawning breast cancer subtype; DESTINY-Breast04 has revealed exciting positive results that trastuzumab deruxtecan almost doubled the progression-free survival in advanced HER2-Low breast cancer patients compared to the physician’s choice [7]. Further exploration of novel ADCs and other anti-HER2 drugs in early-stage HER2-Low breast cancer (ClinicalTrials.gov identifier: T-Dxd-NCT04553770; pyrotinib-NCT05165225) is underway, encouraged by the great success of improving the prognosis in a metastatic setting.

However, there is still vagueness in evaluating HER2-Low status [8]. Traditional methods of HER2 examination, IHC and FISH, showed instability in assessing the low range of HER2 expression [9]. In addition, technical and preanalytical factors may also affect the accuracy of HER2 testing. In clinical practice, preoperative core needle biopsy (CNB) is the standard procedure for breast cancer diagnosis and neoadjuvant regimen decision [10,11] due to its high accuracy similar to that of surgical excision specimen (SES) samples in histology and estrogen receptor (ER), progesterone receptor (PR) and HER2 status determination [12,13]. Nevertheless, owing to the relatively small sample size and tumor heterogeneity, there is inevitable discordance in biomarker assessment between CNB and SES samples. In the pre-ADC era, HER2 status in breast cancer was routinely taken as a dichotomous variable to evaluate its positivity or negativity but not as a quantitative variable. Thus, there were lower discriminatory capabilities in distinguishing low HER2 expression from no expression (IHC 0), especially when restricted by the small tissue size of CNB samples [14]. The inaccuracy in the low range of HER2 expression seems to be clinically acceptable and has no impact on treatment decisions in current daily practice [15,16]. However, it may prevent some HER2-Low patients from receiving effective anti-HER2 ADC therapy. Several studies have revealed the intratumoral heterogeneity of HER2 status, especially in HER2 2+ cases [17]. In addition, Miglietta et al. demonstrated an evolution of HER2 expression between paired CNB and SES samples in breast cancer patients who received neoadjuvant treatment [18]. We still wondered about the impact of CNB on the examination of low HER2 expression to determine whether it is a proper method for HER2-Low diagnosis.

Herein, we aimed to evaluate the concordance rate of HER2-Low status between CNB and SES samples, especially in HER2 negative tumors, in early-stage invasive breast cancer patients and to investigate its association with clinicopathological features.

## 2. Materials and Methods

### 2.1. Study Population

Consecutive early-stage breast cancer patients treated at the Comprehensive Breast Health Center, Ruijin Hospital between January 2009 and March 2022 with paired CNB and SES samples were retrospectively reviewed. The inclusion criteria were invasive breast cancer, female sex and complete IHC and FISH results for both CNB and SES samples. Synchronous and metachronous bilateral breast cancer patients, multifocal breast cancer patients, patients who received neoadjuvant therapy and those with de novo stage IV disease were excluded (Figure 1). The clinicopathological data of all enrolled patients were retrospectively retrieved from the Shanghai Jiao Tong University Breast Cancer Database (SJTU-BCDB). This study was reviewed and approved by the independent Ethical Committees of Ruijin Hospital, Shanghai Jiao Tong University School of Medicine.

### 2.2. Histopathological Evaluation and HER2 Testing Algorithms

Histopathological examination of CNB and SES samples was independently performed at the Department of Pathology, Ruijin Hospital independently. A 14-gauge needle was used in a preoperative biopsy by experienced breast-specific surgeons. At least 4 pieces of tumor tissue (0.1 × 1.0 cm per piece) were collected from each patient. The criteria for ER, PR and Ki67 IHC evaluation were adopted according to the latest American Society of Clinical Oncology (ASCO)/College of American Pathologists (CAP) guidelines [19], and the procedure has been described in our previous studies [20,21].

The algorithms for HER2 testing were to first test HER2 expression by IHC and to perform FISH using a HER2/CEP17 dual probe for IHC 2+ patients [19]. HER2 IHC staining was performed on 4-μm slices of formalin-fixed paraffin-embedded (FFPE) specimens of invasive carcinoma using 4B5 (Roche, Switzerland) as the primary antibody against HER2 protein. The results were reported as HER2-Positive if the sample was IHC 3+ or IHC 2+ and FISH-positive (FISH+, a dual-probe HER2/CEP17 ratio of ≥2.0 with an average HER2 copy number ≥ 4.0 signals/cell, or a dual-probe HER2/CEP17 ratio of <2.0 with an average HER2 copy number ≥ 6.0 signals/cell). HER2-Low was defined as IHC 1+ or IHC2+ and FISH-negative (FISH-, a dual-probe HER2/CEP17 ratio of <2.0 with an average HER2 copy number < 6.0 signals/cell, or a dual-probe HER2/CEP17 ratio of ≥2.0 with an average HER2 copy number < 4.0 signals/cell) [22]. HER2-0 referred to HER2 IHC 0. All histological and biological examination results were confirmed according to the 2018 ASCO/CAP guidelines [23] by at least two experienced pathologists, and an extra review was carried out if there was a contradictory result.

According to the 2013 St Gallen International Expert Consensus, all tumors were divided into five molecular phenotypes: Luminal-A (ER positive, PR ≥ 20% positive, and Ki67 < 20%); Luminal-B/HER2-Negative (ER positive, PR < 20% positive or Ki67 ≥ 20%, and HER2 negative); Luminal-B/HER2-Positive (ER and/or PR positive and HER2 positive); HER2-amplified (ER and PR negative and HER2 positive); and triple-negative breast cancer (TNBC, ER, PR, and HER2 negative) [24].

### 2.3. Statistical Analysis

The HER2 status concordance rate was analyzed in all enrolled patients by using the Kappa test, and Kappa values < 0.2, 0.2–0.4, 0.4–0.6 and > 0.6 were considered poor, fair, moderate and good agreement, respectively. The HER2 discordance between paired CNB and SES samples was graphically reported by building Sankey diagrams. The clinicopathological features were compared in HER2-0 and HER2-Low patients with CNB samples according to HER2 discordance and HER2 status change by using the univariate chi-square test and multinomial logistic regression, reporting the odds ratio (OR) with 95% confidence interval (CI). Statistical analysis and image construction were performed using IBM SPSS version 25 (SPSS, Inc., Chicago, IL, USA) and GraphPad Prism version 8.0 (GraphPad Software, San Diego, CA, USA). A two-sided *p* value of <0.05 was considered statistically significant.

## 3. Results

### 3.1. Patient Cohorts and Baseline Characteristics

A total of 5610 early breast cancer patients were included in the analysis: 1066 patients with HER2-0, 3209 with HER2-Low, 1335 with HER2-Positive CNB findings, 909 patients with HER2-0, 3320 with HER2-Low and 1381 with HER2-Positive SES findings (Figure 1). The baseline clinical features and pathological characteristics based on CNB samples are shown in Table 1. The median age was 56.0 (range 22–95) years. The median interval between CNB and radical surgery was 3.9 (range 1–49) days, and 11.5% of the patients underwent radical surgery over a week after CNB. Invasive ductal carcinoma (IDC) was diagnosed in 88.3% of the patients, and 39.9% had grade III tumors. Node-positive disease was found in 38.9% of the patients. The tumor phenotypes on baseline CNB were distributed as follows: 1275 Luminal-A (22.7%), 2222 Luminal-B/HER2-Negative (39.6%), 726 Luminal-B/HER2-Positive (12.9%), 609 HER2-amplified (10.9%) and 778 TNBC (13.9%).

### 3.2. Biomarker Status Changes from CNB to SES Samples

High concordance rates were found with good agreement in evaluating ER (concordance rate 96.10%, Kappa = 0.898, *p* < 0.001, Table 2), PR (concordance rate 92.70%, Kappa = 0.845, *p* < 0.001) and HER2 status as dichotomous variables (concordance rate 98.26%, Kappa = 0.952, *p* < 0.001) between CNB and SES samples. In addition, 18.93% of patients had a variation in the Ki67 category with moderate agreement when 20% was taken as the cutoff value for high and low expression (Kappa = 0.598, *p* < 0.001). Regarding the molecular subtype, we noticed a discordance rate of 18.48% (Kappa = 0.750, *p* < 0.001, Appendix A). Luminal-A tumors demonstrated the highest conversion proportion (N = 488, 38.3%), in which 474 cases changed from Luminal-A to Luminal-B/HER2-Negative tumors.

### 3.3. HER2 Status Changes from CNB to SES Samples

The HER2 status change from CNB to SES samples is shown in Figure 2. The overall discordance rate was 17.63% (Kappa = 0.684, *p* < 0.001, Figure 2A), mostly represented in cases switching from HER2-0 to HER2-Low (N = 530, 9.4%) and from HER2-Low to HER2-0 (N = 387, 6.9%). In detail, among patients with HER2-0 status in the CNB sample, 49.7% (N = 530) experienced a conversion to HER2-Low status in the SES sample, while 387 patients (12.1%) showed a conversion in the opposite direction (from HER2-Low to HER2-0). On the other hand, tumors with HER2-Positive status in the CNB sample showed the highest stability among the three groups, with 2.0% (N = 26) of patients exhibiting a change to either HER2-0 (N = 1) or HER2-Low (N = 25). We further divided the patients into four subgroups according to the IHC and FISH results: HER2 0, HER2 1+, HER2 2+/FISH- and HER2 3+ or 2+/FISH+. The overall discordance rate of HER2 expression was 31.64% (Kappa = 0.573, *p* < 0.001, Figure 2B).

Furthermore, in patients with HER2-Negative tumors (that is, HER2-0 or HER2-Low) tumors in the CNB sample, we found a lower rate of concordance with a fair agreement in evaluating HER2 low status (concordance rate 76.87%, Kappa = 0.372, *p* < 0.001, Figure 3A). Regarding IHC and FISH evolution with three categories (HER2 0, HER2 1+, and HER2 2+/FISH-), the discordance rate in HER2-Negative patients was 59.09% (Kappa = 0.378, *p* < 0.001, Figure 3B). In patients with HER2 1+ tumors in the CNB sample, 16.7% (N = 338) of them changed to HER2 0, and 28.2% (N = 569) changed to HER2 2+/FISH- when re-testing HER2 expression in the SES sample. In addition, 399 HER2 0 and 191 HER2 2+/FISH- patients changed to HER2 1+ in the SES sample.

### 3.4. Clinicopathological Features Associated with HER2 Status Change

Next, we evaluated the association between HER2 status discordance and clinicopathological features in 4275 HER2-Negative patients (HER2-0 and HER2-Low). A total of 989 (23.1%) patients had a discordant HER2 status between CNB and SES samples (Appendix A). ER (*p* < 0.001), PR (*p* < 0.001), Ki67 (*p* < 0.001) and molecular subtype (*p* < 0.001) were differentially distributed between patients with concordant and discordant HER2 statuses between CNB and SES samples. Multinomial logistic regression demonstrated that the overall distributions of ER (*p* = 0.024, Figure 4), Ki67 (*p* < 0.001), and molecular subtype (*p* < 0.001) were significantly associated with HER2 discordance. ER negativity (OR 1.23, 95% CI 1.03–1.49, *p* = 0.024) and Ki67 ≥ 20% (OR 1.33, 95% CI 1.14–1.55, *p* < 0.001) were independently associated with a high probability of HER2 discordance rate. Compared to TNBC, Luminal-A (OR 0.63, CI 0.51–0.79, *p* < 0.001) and Luminal-B/HER2-Negative (OR 0.78, CI 0.64–0.94, *p* = 0.010) breast cancers were less likely experience HER2 status discordance between CNB and SES specimens.

## 4. Discussion

To our knowledge, this is the first report focused on the accuracy of CNB samples in evaluating HER2-Low status in a large cohort of treatment-naïve early breast cancer patients, which demonstrated a high discordance rate of 23.13% in HER2-Negative tumors. The proportion of tumors with HER2-Low status was slightly higher in the SES samples than in the paired CNB samples. Almost half of HER2-0 tumors in the CNB samples were found to be HER2-Low tumors in the SES samples, indicating that retesting HER2 status in SES samples is needed for patients classified as HER2-0 by CNB for guiding further novel anti-HER2 ADC treatment. Additionally, high agreements were observed in ER, PR and Ki67 status evaluation.

HER2-Low breast cancer is a newly raised entity not only because of its potential clinical benefits but also because of its dissimilar biological characteristics compared with HER2-0 and HER2-Positive breast cancers. Despite the failure of traditional monoclonal drugs to improve the outcomes of HER2-Low patients in pivotal clinical trials [25], the recent DESTINY-Breast 04 trial has demonstrated HER2-Low breast cancer as a distinct clinical subgroup that can be targeted by novel ADCs treatment [7,26]. Continuous attempts to transfer this experimental scenario to the early breast cancer are in progress. In this context, our study anticipated the forthcoming and imperative need to identify the proper patients who may obtain access to novel HER2-targeted treatment in the new ADC era [27]. In current practice, HER2 status of early breast cancer is routinely tested in CNB samples to guide adjuvant anti-HER2 therapy. However, we found that approximately half of the patients identified as HER2-0 by CNB indeed had a certain degree of HER2 expression in the subsequent surgery samples, indicating that retesting HER2 status, especially in HER2-Negative tumors, is warranted to guide further anti-HER2 ADC treatment.

In our study, we found that the HER2 status determined by CNB, including HER2 0, 1+ and 2+/FISH-, was not accurate. Many challenges exist in precise HER2 low status evaluation, including sample preparation, proper sample selection, antibodies or assays and explanation of results. Furthermore, intratumoral heterogeneity may also cause inaccurate HER2-Low status detection. HER2 heterogeneity has been well described in the literature and can be found in up to 34% of breast tumors [28]. By using the PAM50 test, Agostinetto et al. and Schettini et al. demonstrated that HER2-Low breast cancers are composed of heterogeneous clusters of tumor cells, including 50.8–56.9% Luminal-A, 22.8–28.8% Luminal-B, 13.3–17.7% basal-like and even 3.5–3.6% HER2-enriched tumors [29,30]. Recently, Zhang et al. also confirmed the heterogeneity in HER2-Low breast cancers by using another two similar microarray-based genomic profile analyses, MammaPrint and BluePrint [17]. In addition, the technical aspects of HER2 testing methods are likely a major reason for the HER2 discrepancy [8,31]. According to the current guidelines, the IHC/FISH test is a reliable tool to differentiate HER2-Positive tumors from HER2-Negative tumors [32], while the lower boundary of HER2-Low diagnosis seems to be more confusing. As recommended in the 2018 ASCO/CAP guidelines, the proportion of incomplete and barely perceptible IHC-stained tumor cells was the only way to differentiate IHC 0 and IHC 1+ cases without any auxiliary test [23]. Other technical factors, such as the antibody used in the IHC assay [33] and poor agreement of the IHC test among pathologists in differentiating HER2-Low status [34], could impact HER2-Low diagnosis as well. A gold standard of the laboratory protocol and optimal assay to stratify cases in the low HER2 expression range is lacking. Importantly, the method of sampling also has a great influence on the HER2 expression test. CNB is a minimally invasive preoperative examination for early diagnosis and (neo)adjuvant strategy decision making, but the relatively small amount of biopsy tissue available for examination is an inevitable restriction for an accurate result [31]. Although previous research has demonstrated the high accuracy of ER, PR, Ki67 and HER2 as dichotomous variables (Appendix A), our current study showed a relatively lower concordance rate in HER2-Low status diagnosis by CNB; thus, retesting HER2 status with a larger sample size after radical surgery could provide a more precise diagnosis, especially for HER2-Low status detection. Technical variations and preanalytical factors may be another cause of the poor reproducibility of CNB in HER2-Low diagnosis [13,14], which may explain why we found a higher rate of discrepancy between CNB and SES in IHC 1+ tumors than in IHC 2+/FISH- tumors.

Another point deserving further discussion is the factors impacting HER2 status discordance. In the multivariate logistic regression model, we found that patients with ER-negative, PR-negative, Ki67 ≥ 20% and TNBC tumors were significantly more likely to experience HER2 discordance between CNB and SES samples. More HER2-discordant patients were found in ER-negative and TNBC tumors after adjusting for clinicopathological parameters, indicating the possible crucial role of the ER signaling pathway in influencing HER2-Low breast cancer [35,36,37]. Existing data revealed that HER2-Low status was more frequently found in ER-positive breast cancers [2,6,29], and its positive relationship with the ER expression level was verified in a recently published study from our center [38]. Herein, our findings strengthened the hypothesis that ER-positive tumors with low expression of HER2 were more likely to be profiled as the luminal intrinsic subtype by the PAM50 test and seemed to be a biological entity distinct from tumors with no HER2 expression [6,29,30].

Our work has several strengths. To our knowledge, this is the first study to evaluate the concordance rate of HER2-Low status between paired CNB and SES samples in a large consecutive cohort of early-stage breast cancer patients. Our study provided convincing clinical evidence on the optimizing HER2-Low diagnosis and gave us new insights into the clinical and biological aspects of this newly identified entity of breast cancer. The main limitation of the current study is the retrospective nature, especially for HER2 status evaluation, which may have led to an unavoidable diagnosis and selection mistakes.

## 5. Conclusions

In conclusion, we found a high discordance rate of HER2 low status between paired CNB and SES samples in early-stage breast cancer patients, which was related to ER and Ki67 status. Approximately half of the HER2 0 patients tested by CNB samples were diagnosed as HER2 low tumors using the surgery sample, indicating that HER2-Low status needs to be retested in the SES specimens, especially for patients with HER2 0 status on CNB, guiding further clinical anti-HER2 ADC therapy. New techniques, such as machine learning and radiomics-based models [39,40], could be applied to more precisely define HER2-low status in different types of tumor samples.

## Figures and Tables

**Figure 1 cancers-14-06200-f001:**
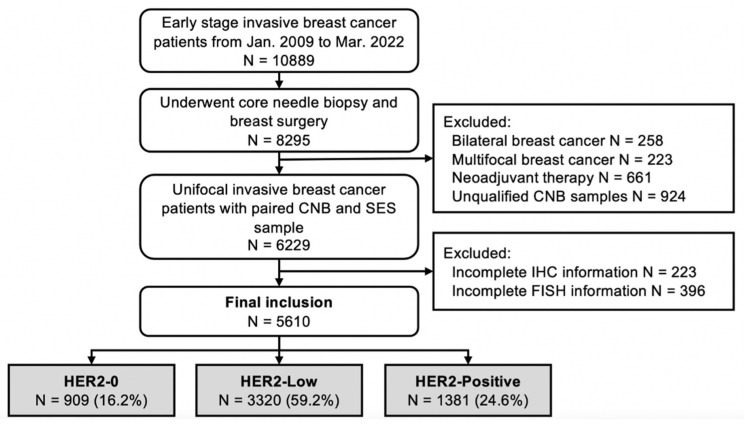
Flowchart of enrolled patients.

**Figure 2 cancers-14-06200-f002:**
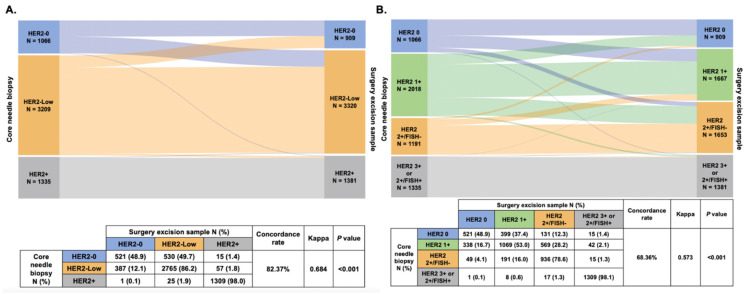
HER2 status change (**A**) and HER2 expression evolution (**B**) from core needle biopsy sample to surgical excision sample. Abbreviations: HER2, human epidermal growth factor receptor-2; FISH, fluorescence in situ hybridization.

**Figure 3 cancers-14-06200-f003:**
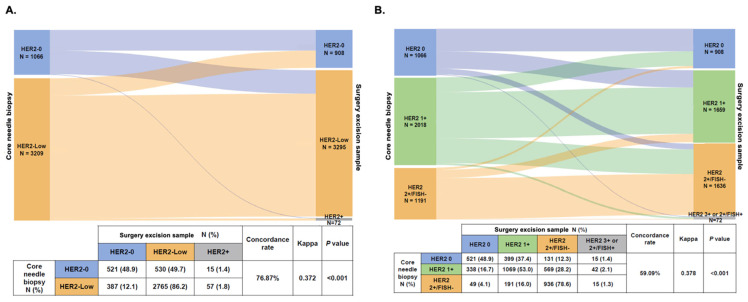
HER2 status change (**A**) and HER2 expression evolution (**B**) from core needle biopsy sample to surgical excision sample in HER2-Negative patients. Abbreviations: HER2, human epidermal growth factor receptor-2; FISH, fluorescence in situ hybridization.

**Figure 4 cancers-14-06200-f004:**
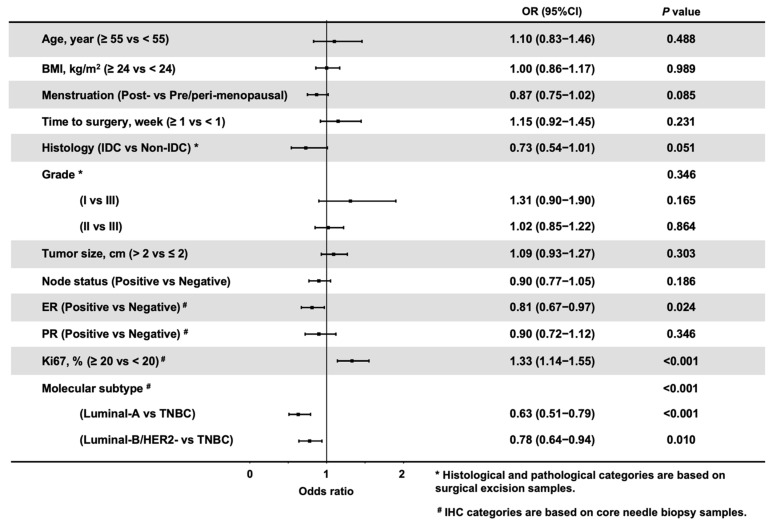
Forest plot of adjusted odds ratios of factors associated with HER2 discordance in HER2-Negative patients. The adjusted estimates were from the logistic model. ^#^ IHC categories are based on core needle biopsy samples. * Histological and pathological categories are based on surgical excision samples. Abbreviations: HER2, human epidermal growth factor receptor-2; BMI, body mass index; IDC, invasive ductal carcinoma; ER, estrogen receptor; PR, progesterone receptor; IHC, immunohistochemistry; TNBC, triple negative breast cancer.

**Table 1 cancers-14-06200-t001:** Baseline clinicopathological features of patients.

Characteristics	Overall Population N = 5610 (%)	HER2-0 ^#^ N = 1066 (%)	HER2-Low ^#^ N = 3209 (%)	HER2-Positive ^#^ N = 1335 (%)	*p* Value
Age, years (median, range)	56.0 (22–95)	57.0 (22–91)	57.0 (24–92)	54.0 (23–95)	<0.001
<55	2543 (45.3)	473 (44.4)	1393 (43.4)	677 (50.7)	
≥55	3067 (54.7)	593 (55.6)	1816 (56.6)	658 (49.3)	
BMI, kg/m^2^					0.019
<24	3464 (61.7)	646 (60.6)	1950 (60.8)	868 (65.0)	
≥24	2146 (38.3)	420 (39.4)	1259 (39.2)	467 (35.0)	
Menstruation					0.626
Pre/peri-menopausal	2034 (36.3)	395 (37.1)	1144 (35.6)	495 (37.1)	
Post-menopausal	3576 (63.7)	671 (62.9)	2065 (64.4)	840 (62.9)	
Time to surgery					0.172
<1 week	4950 (88.4)	950 (89.2)	2809 (87.7)	1191 (89.4)	
≥1 week	645 (11.5)	114 (10.7)	391 (12.2)	140 (10.5)	
NA	15 (0.1)	2 (0.1)	9 (0.1)	4 (0.1)	
Histology *					<0.001
IDC	4952 (88.3)	880 (82.6)	2828 (88.1)	1244 (93.2)	
Non-IDC	658 (11.7)	186 (17.4)	381 (11.9)	91 (6.8)	
Grade *					<0.001
I	209 (3.7)	54 (5.1)	151 (4.7)	4 (0.3)	
II	2748 (49.0)	483 (45.3)	1823 (56.8)	442 (33.1)	
III	2238 (39.9)	427 (40.1)	973 (30.3)	838 (62.8)	
NA	415 (7.4)	102 (9.6)	262 (8.2)	51 (3.8)	
Tumor size, cm					<0.001
≤2	2815 (50.2)	555 (52.1)	1714 (53.4)	546 (40.6)	
>2	2790 (49.7)	510 (47.8)	1493 (46.5)	787 (59.0)	
NA	5 (0.1)	1 (0.1)	2 (0.1)	2 (0.1)	
Nodal status					<0.001
Negative	3414 (60.9)	699 (65.6)	1984 (61.8)	731 (54.8)	
Positive	2183 (38.9)	362 (34.0)	1219 (38.0)	602 (45.1)	
NA	13 (0.2)	5 (0.5)	6 (0.2)	2 (0.1)	
LVI					<0.001
Yes	879 (15.7)	155 (14.5)	438 (13.6)	286 (21.4)	
No	4731 (84.3)	911 (85.5)	2771 (86.4)	1049 (78.6)	
ER ^#^					<0.001
Positive	4152 (74.0)	761 (71.4)	2707 (84.4)	684 (51.2)	
Negative	1458 (26.0)	305 (28.6)	502 (15.6)	651 (48.8)	
PR ^#^					<0.001
Positive	3547 (63.2)	677 (63.5)	2385 (74.3)	485 (36.3)	
Negative	2063 (36.8)	389 (36.5)	824 (25.7)	850 (63.7)	
Ki67 ^#^, %					<0.001
<20	1869 (33.3)	365 (34.2)	1356 (42.3)	148 (11.1)	
≥20	3741 (66.7)	701 (65.8)	1853 (57.7)	1187 (88.9)	
Molecular subtype ^#^					<0.001
Luminal-A	1275 (22.7)	263 (24.7)	1012 (31.5)	0 (0.0)	
Luminal-B/HER2-Negative	2222 (39.6)	503 (47.2)	1719 (53.6)	0 (0.0)	
Luminal-B/HER2-Positive	726 (12.9)	0 (0.0)	0 (0.0)	726 (54.4)	
HER2-amplified	609 (10.9)	0 (0.0)	0 (0.0)	609 (45.6)	
TNBC	778 (13.9)	300 (28.1)	478 (14.9)	0 (0.0)	

^#^ IHC categories are based on core needle biopsy samples. * Histological and pathological categories are based on surgical excision samples. Abbreviations: CNB, core needle biopsy; HER2, human epidermal growth factor receptor-2; BMI, body mass index; IDC, invasive ductal carcinoma; NA, not available; LVI, lymph vascular invasion; ER, estrogen receptor; PR, progesterone receptor; IHC, immunohistochemistry; TNBC, triple negative breast cancer.

**Table 2 cancers-14-06200-t002:** Concordance rate of biomarker status between CNB and SES lesions.

CNB Lesion	SES Lesion		Concordance Rate	Kappa	*p* Value
	Positive	Negative			
**ER**			96.10%	0.898	<0.001
Positive	4062	90			
Negative	129	1329			
**PR**			92.70%	0.845	<0.001
Positive	3296	251			
Negative	158	1905			
**HER2**			98.26%	0.952	<0.001
Positive	1335	26			
Negative ^a^	72	4203			
**Ki67, %**	≥20	<20	81.07%	0.598	<0.001
<20	309	1560			
≥20	2988	753			

^a^ HER2-Negative refers to IHC 0, 1+, or 2+/FISH-. Abbreviations: CNB, core needle biopsy; SES, surgical excision specimen; ER, estrogen receptor; PR, progesterone receptor; HER2, human epidermal growth factor receptor-2; FISH, fluorescence in situ hybridization.

## Data Availability

The data analyzed in the current study are available from the corresponding authors on reasonable request.

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
