# Peer review of "HER2-Low Status Is Not Accurate in Breast Cancer Core Needle Biopsy Samples: An Analysis of 5610 Consecutive Patients"

_cancers, 2022, doi:10.3390/cancers14246200_

Round 1
Reviewer 1 Report
Authors performed a retrospective analysis of 5610 patients who underwent surgery from January 2009 to March 2022 with paired core needle biopsy (CNB) and surgical excision specimen (SES) samples and concluded that there was very low concordance in HER2-Low status between CNB and SES samples in HER2 low status. These findings further indicated the necessity of retesting HER2 low status at surgery.
Major
Manuscript must be correct for factual inaccuracies. For eg. Line 43: Breast Cancer is most common malignancy in women and not across both the genders. Additionally, authors are using older statistics from 2020.
There are grammatical errors and spelling errors across the manuscript which makes it cumbersome to understand what the authors are trying to convey.
These issues need to be fixed for the manuscript to be peer reviewed for the scientific aspect.
Reviewer 2 Report
Thank you for the chance to review this manuscript.
This year, an other manuscript in this topic was reviewd by me entitled : How reliable are gene expression-based and immunohisto-chemical biomarkers assessed on a core-needle biopsy? A study of paired core-needle biopsies and surgical specimens in early breast cancer, by Hani Saghir1 , Srinivas Veerla1 , Martin Malmberg1,2, Lisa Rydén1,4, Anna Ehinger1,3 , Lao H. Saal1 , Johan Vallon- 6 Christersson1 , Åke Borg1 , Cecilia Hegardt1 , Christer Larsson5 , Alaa Haidar 6 , Ingrid Hedenfalk1 , Niklas Loman1,2 7 and Siker Kimbung* 1
In my oppinion, this manuscript complements well the diagnostic and classification problems in breast cancer patients. The study could be extended to other countiers, to have a larger overview of the problem.
Some comments to the manuscript you can find in the attached file.
I apologize for being late with this review.
Kind regards,
Beata Filip-Psurska

Reviewer 3 Report
The manuscript is well-written and suitable for being published on this journal after some revisions.
Title: HER2‐Low status is not accurate in breast cancer core needle biopsy samples: an analysis of 5610 consecutive patients
Authors: Y. Lu, S. Zhu, Y. Tong, X. Fei, W. Jiang, K. Shen, and X. Chen
In this paper, the Authors investigate the accuracy of core needle biopsy (CNB) in diagnosing HER2‐Low status. They found low concordance rate of HER2‐Low status between CNB and surgical excision specimen (SES) samples in early‐stage HER2‐Negative patients.
I’d kindly ask the Authors to answer some comments listed below but in the complex, I think that the content of this paper is very interesting, original, well written, and surely represents a breakthrough in this research field.
For these reasons, I’d kindly ask the Editors to take into consideration the publication of this article on the Cancers journal.
Comments:
1. Authors in lines 70‐87 of the Introduction and in the Discussion mention some works of the literature. Could they provide, if possible, a Table of the these results showing and discussing them in the Discussion section?
2. In Materials and Methods, lines 95‐96, Authors describe the database used in their work. They say that “patients who received neoadjuvant therapy and those with de novo stage IV disease were excluded”. Why weren’t these patients included in the study? Is this related to lack of statistics or because they are not related to the mentioned clinical techniques?
3. In the Discussion, lines 239‐241, Authors say that HER2‐Low breast cancer is a newly raised entity and has potentialities. Could they also stress more the pro and cons of this technique?
4. In lines 253‐254 Authors say “In our study, we found HER2 status on CNB including HER2 0, 1+, 2+/FISH‐ is not accuracy”. I think it should be “accurate”. In general, I suggest Authors to be careful with adjectives and nouns throughout the text.
5. In the Conclusion of the work, Authors find a high discordance rate of HER2‐Low status between CNB and SES samples. Could it be possible to improve this kind of study and its results, for example for the CNB case, implementing Machine Learning algorithms or radiomic‐based models? Indeed Machine Learning and radiomic‐based models, in the breast cancer studies, were extensively employed to analyze breast images, for example, CESM, ultrasounds, and MR, achieving good performances. Throughout the literature, Authors could cite, for example, these works:
(a) A. Fanizzi et al., “Predicting of Sentinel Lymph Node Status in Breast Cancer Patients with Clinically Negative Nodes: A Validation Study”, Cancers 2021, 13(2), 352.
(b) X. Qiu et al., “Could ultrasound‐based radiomics noninvasively predict axillary lymphnode metastasis in breast cancer?”, J. Ultrasound Med. 39, 1897‐1905 (2020);
(c) S. Bove et al., “A ultrasound‐based radiomic approach to predict the nodal status in clinically negative breast cancer patients”, Sci. Rep. 12, 7914 (2022)
Reviewer 4 Report
This study is a very timely subject for HER2 Breast cancer. I thank you for evaluating this study by using many cases. This result, which showed the low concordance of Her2-low status between core needle biopsy and surgical specimen, is very thoughtful for many oncologists to decide the strategy for neoadjuvant treatment in Her2-low breast cancer.
However, you have to re-check the followings;
1. In 3.1 of the results, you described “open surgery” and “radical surgery”. What is the definition of “open surgery” and “radical surgery”?, and Did all patients in this study have “open surgery” after biopsy first, then have “radial surgery”?
2. In Table 1. There is a variable “Time to surgery”. Did this surgery mean “open surgery” or “radical surgery”?
3. In Table 1. The sub-variables at each variable were shown to be confuse. You had better re-arrange the variables and sub-variables.
4. In a comparison of the total numbers of ER, PR, HER2, and Ki67, the numbers were incorrect between Table 1 and Table 2. For example, in the case of ER-positive, the total number of ER-positive in Table 1 was 4152 based on core needle biopsy. But, the total number of ER-positive in Table 2 was 4191 (=4062 + 129) based on core needle biopsy. That was incorrect. The total numbers of PR, HER2, and Ki67 were also different between Tables 1 and 2. I wonder the reason. I want you to check and explain these results?
Round 2
Reviewer 1 Report
Authors analyzed a large cohort of early stage breast cancer patients and found a discordance rate of low HER2 status patients between paired 378 CNB and SES samples. Overall, authors report with evidence the need for re-testing or apply additional testing matrices for HER2 reported patients to validate an indeed low HER2 sample to prevent misdiagnosing them as HER2 subtype of BC. Authors suggest machine learning methods or radiomics based approaches could be potentially promising to address this issue in addition to re-testing low HER2 cases with more material.
The study is well designed and seeks to identify solutions to answer the discordance in HER2 status in early breast cancer patients and its publication would be beneficial to the scientific community and could potentially help start a discussion.
Reviewer 4 Report
Thank you for your work to correct the draft according to my previous comments.
In the coming years, the exact evluation on the status of HER2 will be very important subject as the treatment by ADC expands in the HER2-positive and HER2-low breast cancer.
I hope you will continue to study this research era about HER2 breast cancer